# PRMT5 and CDK4/6 inhibition result in distinctive patterns of alternative splicing in melanoma

Lok Hang Chan[1,2¤a], Peihan Wang[1,2¤b], Shatha Abuhammad[1,2¤c], Lydia Rui Jia Lim[1,2], Joseph Cursons[3], Karen E. Sheppard[1,2,4‡]*, David L. Goode[1,2‡]*

1 Cancer Research Division, Peter MacCallum Cancer Centre, Melbourne, Victoria, Australia, 2 Sir Peter MacCallum Department of Oncology, The University of Melbourne, Parkville, Victoria, Australia, 3 Monash Biomedicine Discovery Institute, Monash University, Clayton, Victoria, Australia, 4 Department of Biochemistry and Pharmacology, The University of Melbourne, Parkville, Victoria, Australia

¤a Current address: School of Agriculture and Food, The University of Melbourne, Parkville, Victoria, Australia
¤b Current address: CAS Key Laboratory of Genome Sciences & Information, Beijing Institute of Genomics, Chinese Academy of Sciences and China National Center for Bioinformation, Beijing, 100101, China
¤c Current address: Department of Medical Oncology, Dana-Farber Cancer Institute, Harvard Medical School, Boston, MA, United States of America
‡ These authors contributed equally as senior and corresponding authors on this work
* david.goode@petermac.org (DLG); karen.sheppard@petermac.org (KES)

**Data Availability Statement:** The dataset(s) supporting the conclusions of this article are available from the Gene Expression Omnibus (GSE209753) available at https://www.ncbi.nlm.

## Abstract

Drugs targeting cyclin-dependent kinases 4 and 6 (CDK4/6) are promising new treatments for melanoma and other solid malignancies. In studies on CDK4/6 inhibitor resistance, protein arginine methyltransferase 5 (PRMT5) regulation of alternative splicing was shown to be an important downstream component of the CDK4/6 pathway. However, the full effects of inhibition of CDK4/6 on splicing events in melanoma and the extent to which they are dependent on PRMT5 has not been established. We performed full-length mRNA sequencing on CHL1 and A375 melanoma cell lines treated with the CDK4/6 inhibitor palbociclib and the PRMT5 inhibitor GSK3326595 and analysed data for differential gene expression and differential pre-mRNA splicing induced by these agents. Changes in gene expression and RNA splicing were more extensive under PRMT5 inhibition than under CDK4/6 inhibition. Although PRMT5 inhibition and CDK4/6 inhibition induced common RNA splicing events and gene expression profiles, the majority of events induced by CDK4/6 inhibition were distinct. Our findings indicate CDK4/6 has the ability to regulate alternative splicing in a manner that is distinct from PRMT5 inhibition, resulting in divergent changes in gene expression under each therapy.

## Background

Cyclin-dependent kinases 4 and 6 (CDK4/6) play an important role in regulating the cell cycle. CDK4/6 when complexed with cyclin D are activated, phosphorylating and inhibiting the retinoblastoma tumour suppressor protein (RB) to ultimately relieve suppression of the

nih.gov/geo/query/acc.cgi?acc=GSE209753. Source code and processed datasets from HTSeq, edgeR, rMATS and EGSEA are available at http://github.com/lokhangc/Melanoma-RNA-Seq-analysis.

**Funding:** This work was supported by the Peter MacCallum Cancer Foundation, in addition to Cancer Council Victoria Grant 1108149 and National Health and Medical Research Council of Australia Grant 1042986 to KES and Victorian Cancer Agency fellowship MCRF17005 to DLG. LHC, LRJL and SA were awarded doctoral scholarships from The University of Melbourne. SA received additional support from Cancer Therapeutics CRC. These funding bodies had no involvement in the design of the study, nor in the collection, analysis and interpretation of data, and were not involved in the writing of this manuscript.

**Competing interests:** The authors declare that they have no competing interests.

transcriptional factor E2F and drive cells into S phase [1]. The CDK4/6 pathway is hyperactivated in around 90% of melanoma cases [2] mainly due to the loss of the tumour suppressor p16 INK4A [3], a direct inhibitor of CDK4/6 activity. Thus, there is a strong push to develop ways to pharmacologically inhibit the CDK4/6 pathway in melanoma and other cancers [2,4,5].

In addition to activating the tumour suppressor RB to halt cell cycle progression, CDK4/6 inhibition (CDK4/6i) also affects numerous other processes through phosphorylation of a diverse array of proteins [6]. One of these proteins is MEP50 [7], a transcriptional co-regulator that modulates protein arginine methyltransferase 5 (PRMT5) substrate specificity and methylation efficiency [8]. PRMT5 regulates many cellular processes and is a major regulator of alternative pre-mRNA splicing [9,10]. Recently in melanoma cell lines, inhibition of CDK4/6 with palbociclib was shown to alter pre-mRNA splicing by modulating PRMT5 activity. Importantly, loss of the ability of CDK4/6 inhibitors to regulate PRMT5-induced alternative splicing conferred resistance to these inhibitors, providing evidence of the importance of alternative splicing in the mechanism of action of CDK4/6 inhibitors [11].

Alternative splicing enables rapid diversification and modification of transcriptomes by altering exon usage to create multiple splice isoforms from the same gene [12,13]. Examples include change of exon boundaries through the use of alternative splice sites, the inclusion or exclusion of specific exons, and retention of introns. Alternative splicing is important in cancer, by creating profound changes to the coding sequence of a transcript and allowing tumour cells to quickly adapt to changes in the intra- and extra-cellular environments [12–14].

The global effects of CDK4/6i on alternative splicing in melanoma and to what extent they depend on PRMT5 are not known. The aim of this study was to identify alternative splice events induced by the CDK4/6 inhibitor palbociclib and assess their overlap with alternative splice events induced by the PRMT5 inhibitor, GSK3326595. Paired-end RNA sequencing was employed to identify alternative splicing events in cells from the CHL1 and A375 melanoma cell line in response to palbociclib and GSK3326595. Although many genes were differentially spliced under both CDK4/6 inhibition and PRMT5 inhibition (PRMT5i), the specific splicing events and the sets of differentially expressed genes varied greatly between each treatment. Our findings shed light on the role of PRMT5 in mediating changes to alternative splicing in response to CDK4/6 inhibition in melanoma.

## Methods

### Cell culture

CHL1 (BRAF$^{wt}$/NRAS$^{wt}$/p53$^{mut}$) and A375 (BRAF$^{V660E}$/p53$^{wt}$) cell lines (were obtained from American Type Culture Collection. A375 cell line were grown in RPMI-1640 Media (Gibco) and CHL1 cell line were grown in DMEM (Gibco). Media were supplemented with 10% (vol/vol) fetal bovine serum (FBS) (Gibco), 2 mM GlutaMAX-I L-alanyl-L-glutamine dipeptide (Gibco), 100 μg/mL penicillin/streptomycin (Gibco) and buffered with 4-(2-hydroxyethyl)-1-piperazineethanesulfonic acid (HEPES). Cell lines were regularly tested for mycoplasma, and the identities of cell lines were confirmed by short tandem repeat (STR) analysis.

Melanoma cells were treated with either the CDK4/6 inhibitor palbociclib (1 μM) or the PRMT5 inhibitor GSK3326595 (500 nM). Four different culture conditions with two replicates each were grown: (i) dimethyl sulfoxide (DMSO) alone added for either 72 hours (control_72hr) or 6 days (control_6d), (ii) 1 μM palbociclib added for 72 hours (CDK4/6i_72hr), (iii) 1 μM palbociclib added for 6 days (CDK4/6i_6d) and (iv) 500 nM GSK3326595 added for 72 hours (PRMT5i_72hr).

## Pharmacological inhibitors

All inhibitors were dissolved in DMSO (Sigma-Aldrich). Palbociclib was purchased from SelleckChem and GSK3326595 from SYNthesis Med Chem.

## Dose-response assays

Dose-response assays were carried out in 96-well plates with five technical repeats per treatment concentration. Drugs were serially diluted in complete RPMI-1640 media to obtain required 10-step concentrations. After the designated treatment length, cells were fixed with 100% (v/v) methanol, then incubated in 2N HCl containing 0.5% (v/v) Triton X-100 (20–30 mins), followed by 0.1 M $Na_2B_4O_7.10H_2O$ (pH 8.5, 10 mins), then propidium iodide (1 µg/mL in PBS) at low light conditions (5 mins). Nuclei (cell number) were then counted. For the analysis, cell numbers were normalized to both a DMSO vehicle control and to the number of nuclei prior to treatment (day zero), obtained through a parallel plate fixed at time of treatment. Using the GraphPad Prism 7 software, $\log_{10}$[inhibitor] vs. response curves were generated and growth inhibition (GI) values determined. From the dose-response assays, obtained GI75–90 values (drug concentrations required to inhibit cell proliferation by 75–90%) were extrapolated to determine the following drug concentrations used in all subsequent experiments, 1 µM Palbociclib and 500 nM GSK3326595.

## RNA isolation

Following treatment, total RNA was isolated using the RNeasy Mini kit (QIAGEN) following the manufacturer's instructions. RNA concentration and purity were determined by a Nano-Drop ND1000 spectrophotometer (Thermo Scientific).

## Alternative MDM4 splicing analysis by PCR

Complementary DNA (cDNA) was generated from 1 µg of total RNA using the High Capacity cDNA Reverse Transcription Kit (Life Technologies) following manufactures instructions. A volume of cDNA equivalent to 30 ng of total RNA was used and 1 µg MDM4-specific primers. Reactions were performed using a S1000 Thermal cycler (Bio-RAD) using the following thermocycling conditions: 95°C for 2 minutes, 27 cycles of 95°C for 45 seconds, 58°C for 30 seconds, 72°C for 40 seconds, and finally 72°C for 5 minutes. Products were visualized on a 2% agarose gel.

## Sequence alignment and gene count

Approximately 1 µg of RNA was used for library preparation according to standard protocols (TruSeq RNA; Illumina). Briefly, poly-A mRNA was purified using poly-T magnetic beads, fragmented using divalent cations under elevated temperature, and reverse transcribed to cDNA with random primers. Indexed adaptors were then ligated, and the library was amplified. Six indexed samples were pooled in a single lane of a NextSeq500 (Illumina) flowcell to generate ∼30 million paired-end 150-bp reads per sample. A total of 28–53 million reads (median 44.4 million) were obtained per sample [S1 Table]. Read quality was assessed using FastQC v0.11.6. Reads were aligned to the GRCh37.73 (hg19) version of the human genome using Hisat 2.0.4. Binary alignment map (BAM) files were sorted and index using Picard v2.17.3 and samtools v1.18. Aligned reads in the BAM files and the GRCh37 GTF annotation file from Ensembl were input to HTSeq v0.13.5 with default settings for read quantification of genes.

## Differential gene expression analysis

The gene count matrix output from HTSeq was used as the input of differential gene expression analysis. The analysis was performed with edgeR (v. 3.32.1) following the procedures and steps described in package documentation. In brief, lowly expressed genes across samples were removed and the expression of the remaining genes was then normalised with the Trimmed Mean of M-values (TMM) method, which calculates the relative gene expression of a selected sample. The dispersion of genes was further estimated and fitted into a negative binomial model together with a contrast matrix to determine the differentially expressed genes (DEGs) between treatments. Genes were declared significantly differentially expressed with a false discovery rate (FDR) < 0.05. Treatments with two replicates each were compared. Single contrast analysis comparing treatment and the control was performed: {CDK4/6i_72hr vs Control}, {CDK4/6i_6d vs Control} and {PRMT5i_72hr vs Control}. Multi-contrast comparison further performed compared the list of DEGs between: {{CDK4/6i_72hr vs Control} vs {CDK4/6i_6d vs Control}} and {{CDK4/6i_6d vs Control} vs {PRMT5i_72hr vs Control}} [S1 Fig]. For A375, CDK4/6i_72hr and PRMT5i_72hr samples were compared to the control_72hr samples, while the CDK4/6i_6d was compared to the control_6d samples.

## Differential alternative splicing analysis

BAM files and the Genome Reference Consortium Human Build 37 (GRCh37) gene transfer format (GTF) annotation file were used as inputs for the identification of splicing events. Differential alternative splicing analysis was performed with the replicate multivariate analysis of transcript splicing (rMATS) algorithm (v.4.1.0) to detect five categories of splicing events including skipped exon (SE), intron retention (RI), mutually exclusive exons (MXE), alternative 5' splice site (A5SS) and alternative 3' splice site (A3SS) [Fig 2A]. An anchor length of 8 was used for a more conserved prediction. The variable read length mapping and novel splice site detection options were enabled to maximise the prediction of splice events. To determine inclusion and skipped levels of splicing events, read length was restricted to 149 bp. Most reads (~80%) in our data set had a read length of 149 bp. Default settings were used for other non-specified parameters. Only the splicing events defined by junction counts were used for further investigation. A significance threshold of FDR < 0.05 and inclusion level difference > 0.05 were used to define significant differential splicing events. Genes with significant differential splicing events were defined as differential spliced genes for further analysis. Single contrast and multi-contrast comparison were carried out as per the differentially expressed genes [S1 Fig].

## Gene set enrichment analysis on DEGs and differentially spliced genes (DSGs)

DEGs and DSGs were annotated with the "C5" collection with ontology gene sets information from msigdbr (v. 7.4.1). For a direct comparison of the enriched gene sets of DEGs and DSGs, an over-representation analysis (ORA) with the hypergeometric distribution approach was performed with the Ensemble of Gene Set Enrichment Analyses (EGSEA; v. 1.18.1). Gene Ontology (GO) sets belonging to the biological process components were selected to create an annotation object, with the lists of DEGs or DSGs with the annotation object input for ORA. A significance threshold of adjusted p-value < 0.05 was used to determine the significantly enriched gene sets.

## Heatmap and dendrograms

DEGs or DSEs commonly discovered across all treatments were selected to generate heatmaps using ggplot2 (v. 3.4.2). The value of log fold-change were used for DEGs and the value of

inclusion level difference were used for DSEs. Clustering analysis were performed based on the magnitude and direction of the value of log fold-change or inclusion level difference using the average distance calculation. Dendrograms were generated with the ggdendroplot (v. 0.1.0) package using the average distance matrix.

# Results

## Changes to gene expression and splicing during prolonged CDK4/6 inhibition

CHL1 and A375 melanoma cells were subjected to treatment with 500 nM of the PRMT5 inhibitor GSK3326595 for 72 hours or with 1 μM of the CDK4/6 inhibitor palbociclib for either 72 hours or 6 days. At these concentrations Palbociclib and GSK3326595 induce a 75–90% inhibition of cell proliferation [S2 Fig] and a decrease in PRMT5 activity, as indicated by the levels of H4R3 symmetric demethylation [11]. However, it is important to note that complete inhibition of PRMT5 activity is not achieved with either drug [11]. Notably, consistent and detectable changes in alternative splicing were observed with both drugs by the 72-hour time point [11]. Paired-end RNA sequencing reads with a length of 150 bp were generated for all samples. Following read alignment and filtering, with 28–53 million reads per sample successfully mapped to transcripts from the Ensembl hg37 annotation set. The expression levels of genes were found to be highly consistent across samples, with 22,675 to 23,848 genes detected per replicate employing HTseq v0.13.5 [15] [S1 Table].

Selection of the appropriate CDK4/6i treatment length for this study was assessed according to the time-dependent effects of Palbociclib on the two melanoma cell lines. Transcriptome composition after 72 hours and 6 days of inhibition were compared using Principal Component Analysis (PCA). Sample replicates clustered together according to time on treatment [Fig 1A and 1B]. Subsequently, data from each treatment were independently compared against corresponding controls to identify transcripts with differential expression using edgeR [16]. In both CHL1 and A375, the number of differentially expressed genes (DEGs) increased during of CDK4/6i treatment. Following 72 hours of CDK4/6i treatment, CHL1 exhibited a total of 943 DEGs (445 up-regulated/498 down-regulated) which increased to 3300 DEGs (1689 up-regulated/1611 down-regulated) after 6 days of CDK4/6i treatment [S2 Table]. A similar trend was observed in A375, with 3119 DEGs (1710 up-regulated/1409 down-regulated) detected, which increased to 9477 DEGs (4876 up-regulated/4601 down-regulated) after 6 days [Fig 1C]. Notably, there was a high concordance between time points in both CHL1 and A375, with 92.2% and 75.8% of the observed DEGs at 72 hours also differentially expressed after 6 days, respectively [S3 Fig].

In addition to gene expression, CDK4/6i also regulates gene splicing. Therefore, we examined the effect of CDK4/6i on alternative splicing in both melanoma cell lines. To analyse this, we utilized the replicate multivariate analysis of transcript splicing (rMATS) algorithm, which processes RNA sequencing reads from all replicates of CDK4/6 inhibited and control samples. The rMATS software evaluates five major alternative splicing events: alternative 3' splice sites (A3SS), alternative 5' splice sites (A5SS), mutually exclusive exons (MXE), retained introns (RI), and skipped exons (SE) [Fig 3A]. Each sample's splicing events are quantified, and an "inclusion level" is calculated to indicate the frequency of each alternatively spliced transcript relative to the annotated transcript for the corresponding gene [17]. Note that only splicing events involving known annotated exons are considered. A statistical model accounts for splice isoform length and variability between replicates to identify specific splicing events with statistically significant differences in their inclusion levels between conditions. We identified individual splicing events in the 72-hour and 6-day CDK4/6 inhibited samples and quantified the

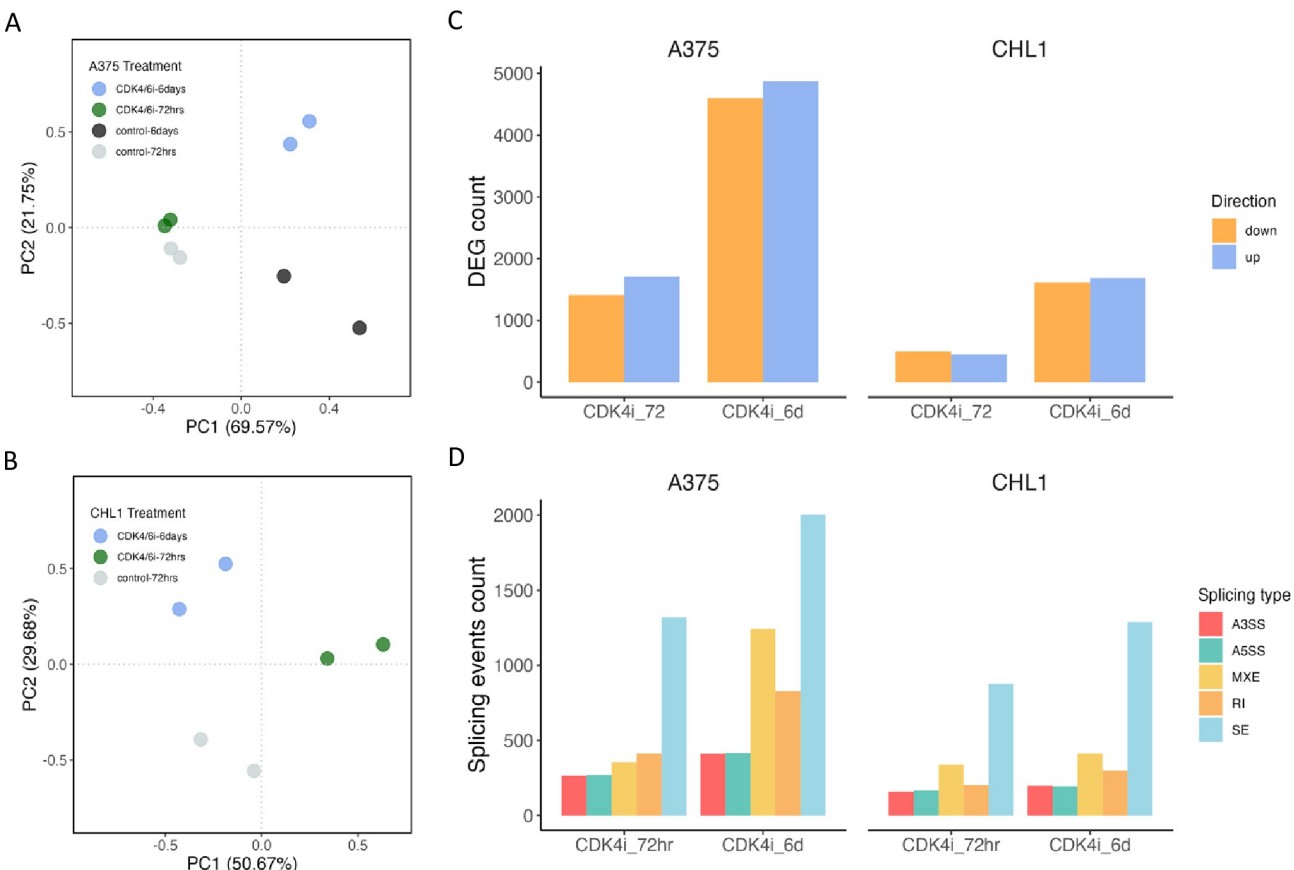

**Fig 1. Prolonged CDK4/6 inhibition response of gene expression and splicing in A375 and CHL1.** PCA plot of CDK4/6 inhibited samples showing clusters between replicates in (**A**) A375 and (**B**) CHL1 cell. (**C**) Number of upregulated and downregulated genes in CDK4/6 inhibited samples in A375 (left) and CHL1 (right) cell. (**D**) Number of differentially spliced events in CDK4/6 inhibited samples in A375 (left) and CHL1 (right) cell.

number of reads supporting each splice isoform. We compared these results to DMSO control samples to identify splicing events with significant changes in frequency upon treatment, using a threshold of FDR <0.05 (Methods). We refer to these observations as "differential spliced events (DSEs)", which indicate alterations in the alternative splicing of mRNAs in response to drug treatment.

Overall, CDK4/6i resulted in a total of 1,741 and 2,621 DSEs in CHL1 and A375 after 72 hours, respectively, increasing by 37.3% and 87.1% to reach 2,391 and 4,904 after 6 days of treatment. These changes were observed across all types of splice events detected by rMATS, with skipped exons (SE) being the most frequently detected event [Fig 1D]. Approximately equal proportions of DSEs were found to be enriched (upregulated) and depleted (downregulated) in all CDK4i treated cells. Specifically, 51% of splice events were upregulated by CDK4/6i at 6 days and 52% at 72 hours in CHL1, while in A375, 49% were upregulated at 6 days and 45% at 72 hours [S4 Fig].

The results indicate prolonged CDK4/6i impacts both gene expression and alternative splicing in CHL1 and A375 melanoma cells. Therefore, the numbers of DEGs and DSEs observed at 72 hours primarily represent an early response to CDK4/6i that is sustained, rather than an immediate-early transcriptional program or stochastic variation in gene expression. The consistent pattern of wide-scale changes in the transcriptome of CHL1 and A375 cells emerges only after prolonged exposure to palbociclib. Consequently, the 6-day CDK4/6i samples were selected for the downstream analysis of the CDK4/6i dependency on PRMT5.

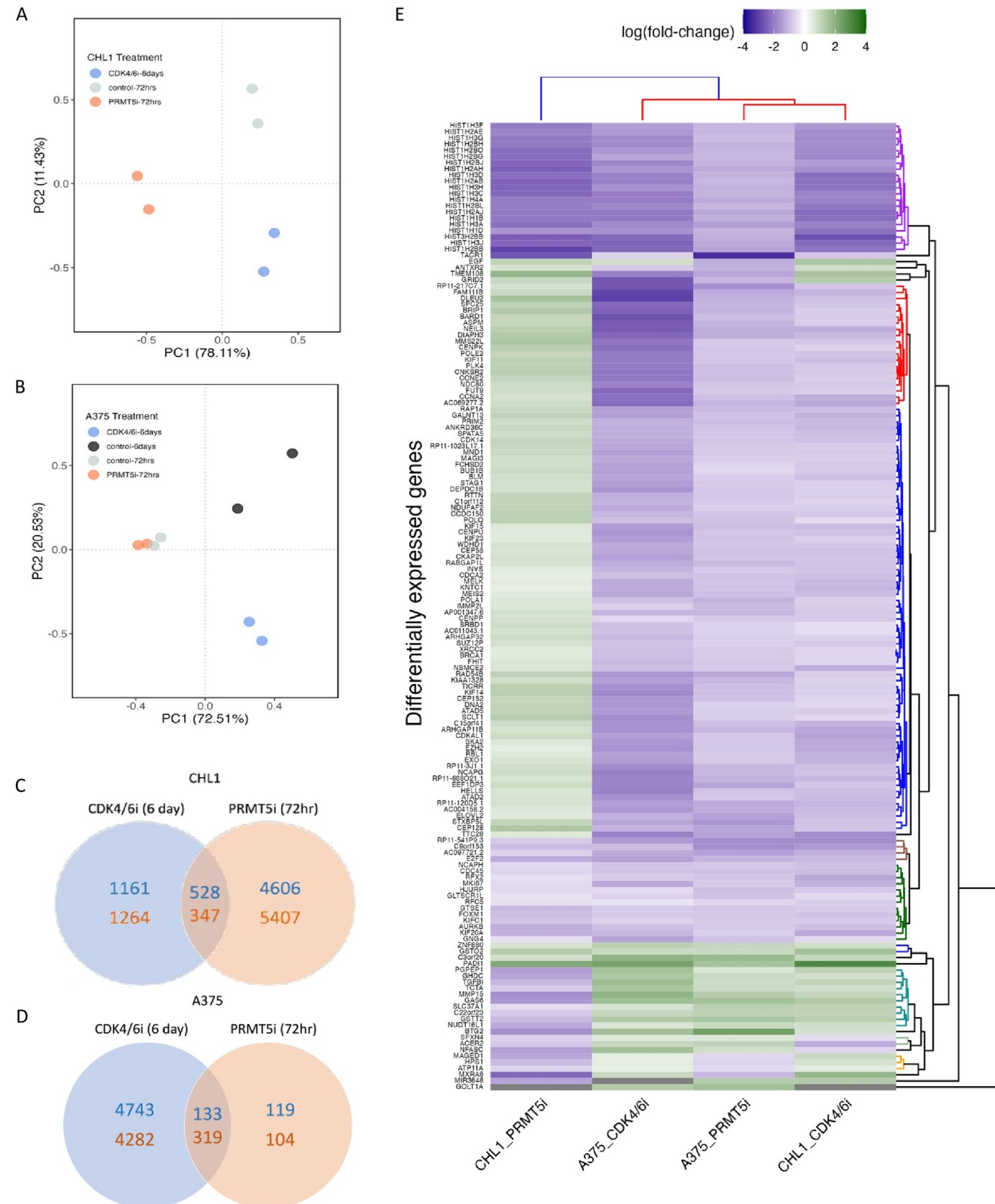

**Fig 2. Differential gene expression in response to CDK4/6 and PRMT5 inhibition.** PCA plot of CDK4/6 and PRMT5 inhibited samples showing clusters between replicates in **(A)** A375 and **(B)** CHL1 cell. Number of upregulated and downregulated genes shared in CDK4/6 and PRMT5 inhibited samples in A375 **(C)** and CHL1 **(D)** cell. **(E)** Fold-change of differentially expressed genes shared across all samples.

## CDK4/6 and PRMT5 inhibition generate divergent changes in gene expression

Although PRMT5 is a protein within the CDK4/6 pathway that regulates various cellular processes, how the global effects of CDK4/6 inhibition in melanoma, relate specifically to PRMT5, are currently not well understood. To investigate, we compared the gene composition of melanoma cells treated with CDK4/6 and PRMT5 inhibitors using PCA, which revealed each treatment reproducibly results in distinct expression profiles across the two cell lines [Fig 2A and 2B]. We then compared DEGs between CDK4/6 and PRMT5 inhibition.

PRMT5 inhibition induced nearly four times more DEGs than CDK4/6 inhibition. Specifically, in PRMT5i treated CHL1 cells, a total of 10,888 DEGs (5134 upregulated/5754 downregulated) were identified [Fig 2C]. This demonstrates that PRMT5i induces more pronounced and immediate changes to the transcriptome of CHL1 cells compared to CDK4/6 inhibition. However, in A375 cells, only 675 DEGs were observed after PRMT5i, which is a marked difference from the DEG numbers in CHL1 [Fig 2D]. This suggested huge variations in proliferative capacity and genetic makeup between the cell lines. Notably, Palbociclib and GSK3326595 appear to induce divergent changes in gene expression in both CHL1 and A375 cells, with each treatment altering the expression of distinct sets of transcripts (Fisher's Exact test, $p<0.05$) [Fig 2C and 2D]. After 72 hours of CDK4/6 there is also low overlap in DEGs [S5 Fig].

To assess the dependency of PRMT5 on CDK4/6 inhibition, we examined the overlap of DEGs between both treatments. In CHL1, a total of 875 DEGs (528 up-regulated/347 down-regulated) were shared between CDK4/6 and PRMT5 inhibition, accounting for only 6.6% of all detected DEGs [Fig 2C]. Similarly, in A375, there were 452 shared DEGs (133 up-regulated/319 down-regulated), representing just 4.4% of all detected DEGs [Fig 2D]. Among these shared DEGs, 157 genes were common to both CHL1 and A375. Of the common DEGs, 39 were down-regulated across all treatments, with histone genes clustering together [Fig 2E]. There were also other important genes regulating cell cycle, such as the cell division cycle (CDC) 45 and E2F transcription factor 2 (E2F2) genes. Interestingly, contrasting responses between the two cell lines under PRMT5 inhibition occurred. Some genes that were down-regulated in A375 under both CDK4/6 and PRMT5 inhibition were in fact up-regulated in PRMT5i-treated CHL1 [Fig 2E]. The cluster analysis further revealed the gene expression response of CHL1 to PRMT5i differs noticeably compared to other treatments [Fig 2E]. Although the two melanoma cell lines tested here exhibited a similar gene expression response under CDK4/6i, they demonstrated a divergent response to PRMT5i.

## CDK4/6 and PRMT5 inhibition produce distinct changes to alternative splicing

Alternative splicing can interfere with gene expression and protein formation, leading to changes in pathway function. Understanding the effects of CDK4/6 inhibition on alternative splicing in melanoma cells and its dependency on PRMT5 is crucial. Therefore, we compared the DSEs detected by rMATS [Fig 3A]. PRMT5i induced a total of 10,843 DSEs in CHL1, which was more than four times higher than that of CDK4/6i with 2,391 DSEs detected [S3 Table]. Similarly, 5,868 DSEs were detected in A375 treated with PRMT5i, which was slightly higher than that of CDK4/6i with 4,904 DSEs detected [Fig 3B]. Nearly half (51%) of the

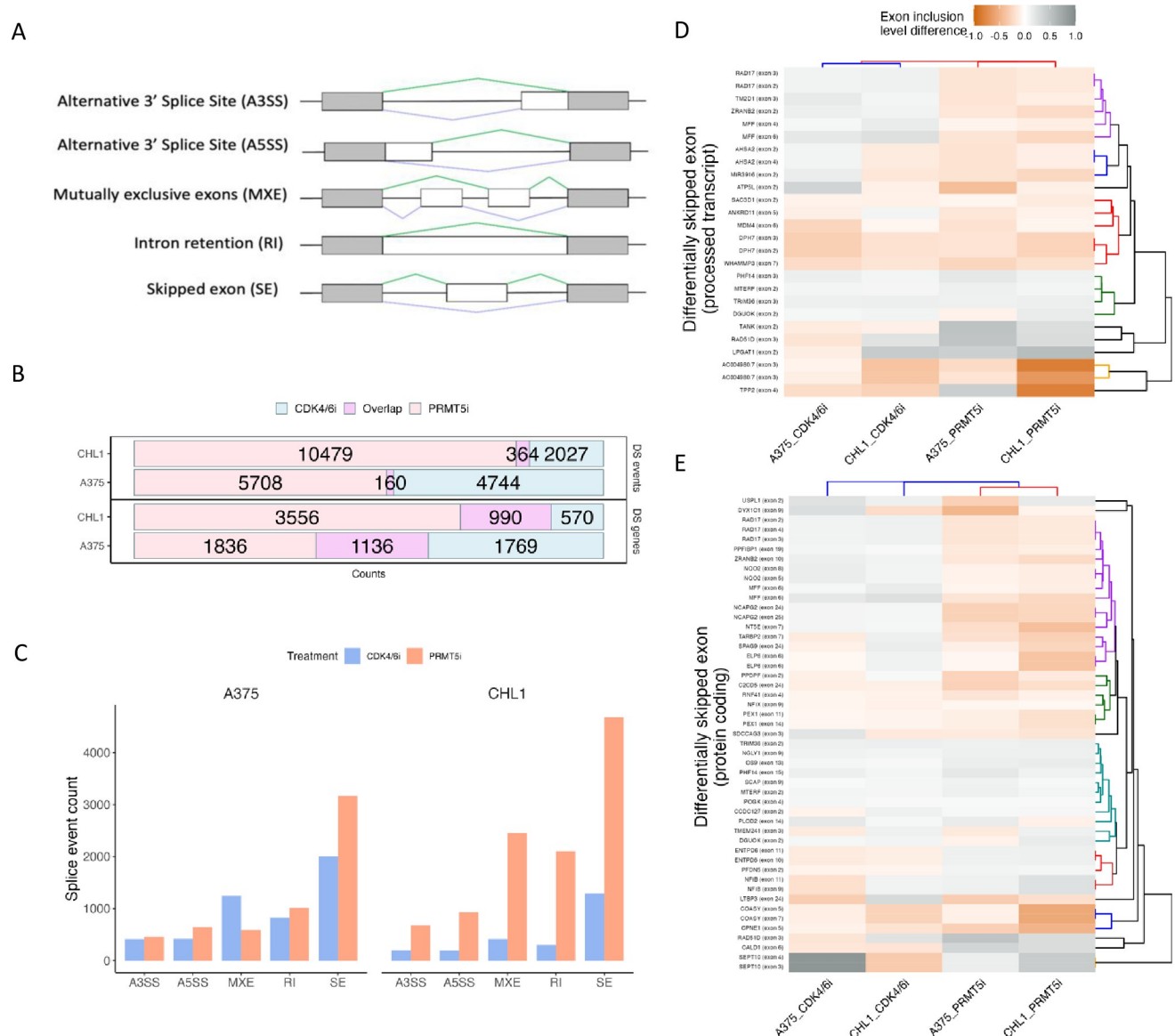

**Fig 3. Differential splicing in response to CDK4/6 and PRMT5 inhibition. (A)** Types of alternative splicing events identified by rMATS. **(B)** Number of differentially spliced events and differentially spliced genes shared in CDK4/6 and PRMT5 inhibited samples. **(C)** Abundance of different types of splice events in CDK4/6 and PRMT5 inhibited samples. Exon inclusion level difference of skipped exons events detected across all samples, related to processed transcript **(D)** and protein coding **(E)**.

PRMT5i-induced DSGs harboured multiple splicing events, whereas this was less for CDK4/6i; 28% at 6 days and 26% at 72 hours [S6 Fig]. This trend was consistent across all types of splice events detected by rMATS, with skipped exons (SE) being the most frequently observed event in all treatment conditions [Fig 3C]. Among all the detected DSEs, 2.8% were shared between CDK4/6 and PRMT5 inhibition in CHL1, which was lower in A375 with 1.5% shared [Fig 3B], with low overlap across all types of DSEs [S7 Fig]. This indicates that the CHL1 cell has a stronger response of alternative splicing under PRMT5i and more of these DSEs detected in CDK4/6i might be dependent on PRMT5. The high proportion of distinct DSEs detected under PRMT5i is consistent with the known functions of PRMT5 as a direct regulator of the spliceosome [1,18,19].

Skipping of exon 6 in MDM4 has been reported to be regulated under CDK4/6i via PRMT5 [11]. DSE with an unadjusted p-value <0.05 captured MDM4 SE events across all samples [Fig 3D], which matched Polymerase Chain Reaction (PCR) data from a previous related study showing suppression of PRMT5 activity leads to decreased MDM4-full length (FL) mRNA and thus protein expression through the skipping of exon 6 in A375 [S8 Fig] [11]. This specific SE event was more frequent in PRMT5i treated CHL1 and A375 cells than in CDK4/6i treated samples [S4 Table]. Overall, SE events were detected in 26 processed transcripts [Fig 3D] and 49 protein coding transcripts [Fig 3E] at the relaxed threshold. Cluster analysis additionally showed PRMT5i treated samples have a similar skipping exon response based on the commonly detected DSEs [Fig 3D and 3E]. Note that even with the relaxed significance threshold of unadjusted p < 0.05, the overlap in DSEs between treatments remained low [S9 Fig].

Overall, these findings highlight the distinctive effects of CDK4/6 and PRMT5 inhibition on alternative splicing in melanoma cells. CHL1 cells show a more pronounced response in terms of differentially spliced events, with MDM4 exon 6 being identified as a common skipped exon event. These results emphasize the importance of considering the specific effects of different inhibitors on alternative splicing and their potential implications for melanoma treatment.

We then associated the DSEs with genes by investigating the differentially spliced genes (DSGs), which were defined as genes harboring one or more DSEs at FDR <0.05. Consistent with the spliced event data, PRMT5i resulted in a higher number of DSGs (4,546) compared to CDK4/6i (1,560) in CHL1, with 19.3% commonly spliced in both treatments. Similarly, PRMT5i and CDK4/6i in A375 resulted in 2,972 and 2,905 DSGs respectively, with 24% commonly spliced [Fig 3B]. Similar degree of overlap in DSGs was seen between 72 hours CDK4/6i and PRMT5i samples [S10 Fig] Across all treatments, 236 DSGs were commonly detected, with genes participating in the cell cycle, including the cell division cycle (CDC) 27 gene and the cyclin dependent kinase (CDK) 16 gene. Since we have observed the MDM4 skipped exon 6 events with the p-value <0.05 threshold, we also compared the DSGs identified with this loosened threshold. There are 47% and 49% of genes being spliced under both treatments in A375 and CHL1 respectively [S9 Fig]. In particular, cell cycle related genes such as CDK2, E2F3, E4F transcription factor 1 (E4F1), MDM2 and MDM4 to name a few, were also being spliced across all treatments. This indicates that both CDK4/6i and PRMT5i have influenced splicing in genes that contribute to the cell cycle.

## The functional impacts of differential expression and differential splicing vary depending on the treatment

Distinct patterns of differential expression and splicing in genes were observed under CDK4/6 and PRMT5 inhibition. To investigate the relationship between differential splicing and gene expression changes, we compared DSGs (FDR <0.05) with DEGs. We found that 71.5% of the DSGs in PRMT5i-treated CHL1 were also identified as DEGs, whereas only 19.3% of the DSGs in CDK4/6i-treated CHL1 were also identified as DEGs. In A375, only 3.5% of the DSGs in PRMT5i-treated cells but 45.8% of the DSGs in CDK4/6i-treated cells were also identified as DEGs [Fig 4A]. The high proportion of DSGs overlapping with DEGs in PRMT5i-treated CHL1 suggests that CHL1 cells are particularly sensitive to PRMT5 inhibition compared to A375 cells. Additionally, the changes in splicing have a high potential to alter gene expression in PRMT5i-treated CHL1.

To further assess the functional implications of the DEGs and DSGs, we performed gene set enrichment analysis using the hypergeometric distribution and the Ensemble of Gene Set Enrichment Analyses (EGSEA) package [20]. We tested for over-representation of the

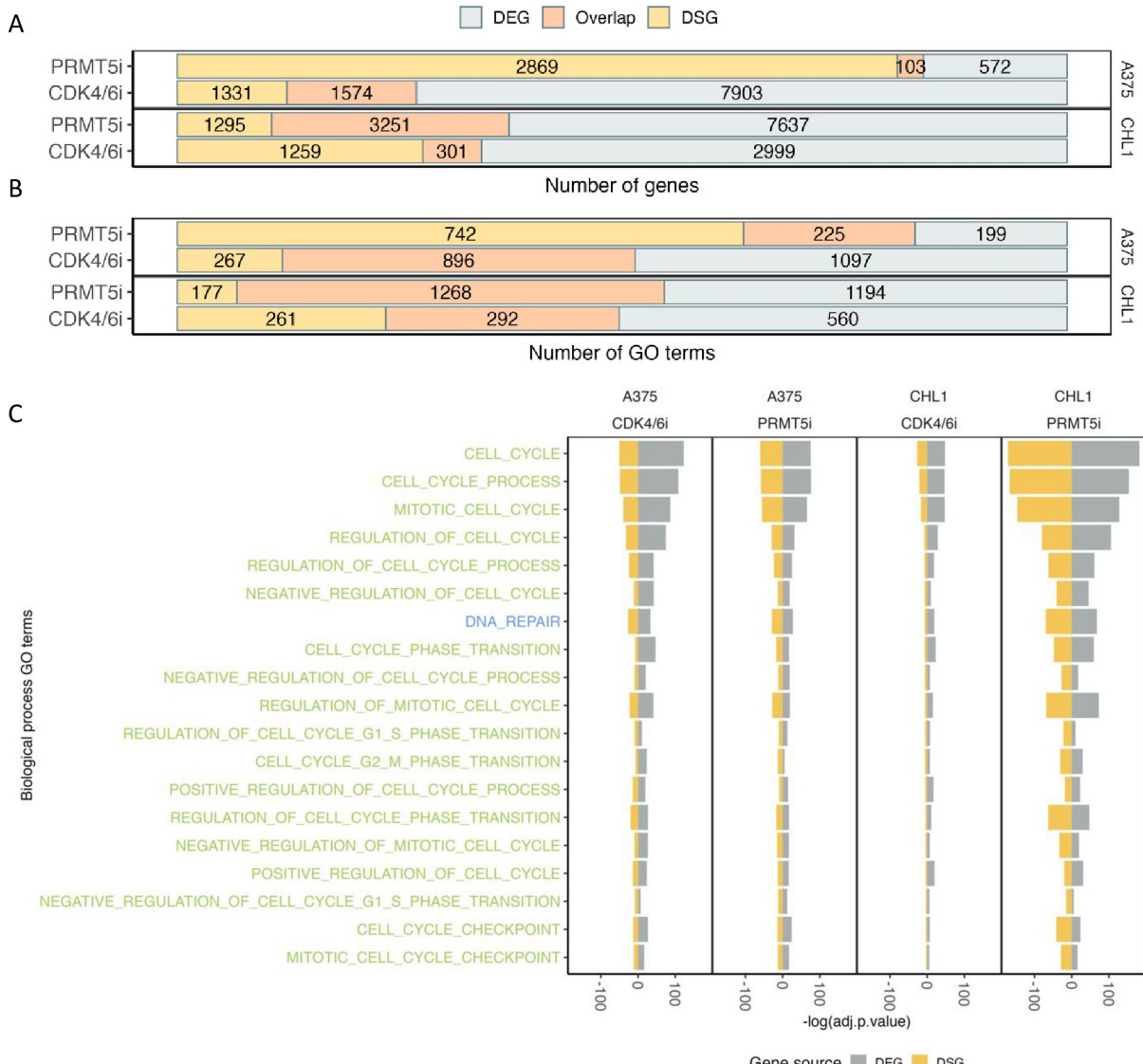

**Fig 4. Functional responses in differentially expressed and spliced genes.** (A) Number of genes detected as differentially expressed and differentially spliced. (B) Number of GO gene sets determined using the differentially expressed and spliced genes. (C) Significance of gene sets detected across all samples and in both differentially expressed and spliced genes relating to cell cycle (green) and DNA repair (blue).

MSigDB C5 collection (Biological Processes from the Gene Ontology database) [21] within each gene list. Altogether, we observed that 1,113 Gene Ontology (GO) terms were enriched in the DEGs or DSGs from CDK4/6i-treated CHL1, with 26% of them being enriched in both categories. In the same cell line, under PRMT5 inhibition, a much higher number of GO terms were enriched, with a total of 2,639 terms, and 48% of them being shared. Interestingly, an opposite pattern was observed in A375, with a higher number of overlapping GO terms reported in the CDK4/6-treated samples [Fig 4B]. Among all the enriched terms, 94 were related to cell cycle, DNA repair, splicing, and translation, which are core processes directly affected by the inhibition of CDK4/6 and PRMT5. In CHL1, there were more GO categories enriched within the DEG list compared to the DSGs in both treatments, which was not the same as A375. The PRMT5i treated CHL1 have the most number of enriched GO terms

among all treatment, with 65 shared in DEGs and DSGs. This indicates more functions being enriched in PRMT5i treated CHL1 [S11 Fig].

Other key biological processes enriched for DEGs and/or DSGs included the regulation of cell death, intracellular signalling, and mitotic processes [S12 Fig]. This indicates that under CDK4/6 inhibition, differential gene expression and differential RNA splicing can impact different sets of transcripts related to cell growth, proliferation, splicing, and translation. The cell cycle biological process has the most significant enrichment across all treatments [S11 Fig]. Genes observed across treatments such as CDC27 and CDK16 in DSGs, as well as E2F2 and epidermal growth factor (EGF) in DEGs, were also observed to be related to cell cycle. We have also observed genes encoding the serine and arginine rich splicing factor (SRSF), RNA binding motif protein (RBM), U2 small nuclear RNA auxiliary factor 2 (U2AF2) and eukaryotic translation inhibition factor (EIF) in the majority of the treatments which were related to gene splicing or translation.

We found 19 GO terms commonly enriched across all samples, as measured by the adjusted p-value. Notably, 18 of these terms were related to cell cycle, with the other being 'DNA repair' [Fig 4C]. Enrichment in DEGs was more significant than that for DSGs, although this significance was similar in DSGs and DEGs in CHL1 treated with PRMT5i. Overall our results illustrate the different responses to treatment of DEGs, DSEs and enriched GO terms between CHL1 and A375. The distinct pattern of DEGs, DSGs and enriched GO terms in CHL1 indicates that the changes in gene expression and splicing events under CDK4/6i were not completely controlled through PRMT5.

## Discussion

While PRMT5 is a known regulator of alternative splicing, the effects of CDK4/6 are less defined. CDK4/6 has been proposed to regulate splicing by modulating PRMT5 activity; however, it is unknown whether CDK4/6 alters splicing through other mechanisms. In this study, we used paired-end sequencing of full-length mRNA transcripts to quantify and compare the effects of CDK4/6 inhibition and PRMT5 inhibition on gene expression and alternative RNA splicing within the CHL1 and A375 melanoma cell lines. Not surprisingly we identified differences in gene expression in response to CDK4/6 inhibition and PRMT5 inhibition across both lines, with PRMT5 inhibition inducing more DEGs in CHL1 but CDK4/6i more DEGs in A375. The majority of DEGs were unique to one treatment, demonstrating that in our model CDK4/6 inhibition and PRMT5 inhibition induce distinct transcriptional responses. Note that although a large number of genes exhibited different responses between treatments and cell lines, a consistent down-regulation of histone formation genes was observed across all treatments. This suggests regulation of histone formation genes under CDK4/6 inhibition were also observed under PRMT5 inhibition.

Interestingly, although there were common alternative splicing events with both treatments, most splicing changes induced by CDK4/6 inhibition were also unique, indicating CDK4/6 inhibition can regulate alternative splicing via mechanisms that are distinct from PRMT5 inhibition. However, the overlap in differentially spliced genes was much more pronounced, with nearly 40% of CDK4/6 inhibitor induced differentially spliced genes were also found to be differentially spliced after PRMT5 inhibition in A375, rising to approximately 70% in CHL1. Thus, we found that while many of the same genes undergo alternative splicing in response to both CDK4/6 and PRMT5 inhibition, those genes were often spliced at different exons, implicating additional factors beyond PRMT5 which may regulate splicing in response to CDK4/6 inhibition.

The functional impact of DEGs and DSGs varied between treatments too. Overall, there was low overlap in the DEGs and DSGs found in the CDK4/6i CHL1 samples. Consistent with

its mode of action CDK4/6 inhibition has a greater impact at the transcriptional level after 6 days of treatment. Results varied with high overlap of DEGs and DSGs from PRMT5i samples from CHL1 but lower overlap in A375 and GO terms relating to cell growth and DNA repair tended to show stronger enrichment within DEGs than DSGs under CDK4/6 inhibition. In contrast, inhibiting PRMT5 had more consistent effects on these processes at the expression and splicing levels in both cell lines.

Our data provide a basis for determining how changes to splicing contribute to the efficacy of CDK4/6 inhibitors and how splicing changes brought about by CDK4/6 inhibition may introduce additional therapeutic vulnerabilities or lead to emergence of resistance mechanisms. Future studies in additional model systems and patient cohorts will refine and highlight the common effects on splicing of CDK4/6 inhibition in melanoma and other solid cancers. Furthermore, the rMATS algorithm applied here detects splice events involving known annotated exons, thus missing alternative splicing which involves cryptic or unannotated exons. Accordingly, development of new computational methods may provide further insights on complex splicing changes in melanoma.

Here we show inhibition of CDK4/6 regulates alternative splicing across the transcriptome in ways that are distinct from PRMT5 inhibition. Our observations remain concordant across two distinctive melanoma cell lines with different origins and genetics. This suggests the impact of CDK4/6 inhibition on splicing is not due to inhibition of PRMT5 alone and raises the possibility that CDK4/6 inhibition may alter the behaviour of PRMT5 or alternative splicing factors to change the use of splice sites and the repertoire of exons targeted for splicing. The distinct patterns of differential splicing and differential expression between the two treatments may explain why combination of CDK4/6i and PRMT5i has synergistic effects in multiple melanoma cell lines [2,11,14].

A number of promising therapies exist to exploit splicing processes in cancer. Targeting splicing in combination with CKD4/6 inhibition is a promising line of investigation for more effective and lasting treatments for melanoma and other cancers. Our work indicates realising the full potential of this strategy will require understanding of the divergent and potentially complementary effects on alternative splicing between CKD4/6 inhibitors and other agents that impact on splicing.

## Supporting information

**S1 Table. Numbers of reads, library size and HTSeq gene count for samples from CHL1 and A375 cells.**
(DOCX)

**S2 Table. Number of differentially expressed genes in CHL1 cells under different treatments.**
(DOCX)

**S3 Table. Numbers of differential splicing events of each type detected by rMATs at each time point post-treatment for CHL1 and A375 cells.**
(DOCX)

**S4 Table. MDM4 skipped exon events of exon number 6 detected by rMATS.**
(DOCX)

**S1 Fig. Schematic of the single contrast analyses (solid lines) comparing each treatment to the control and multi-contrast analyses (dashed lines) comparing differentially expressed**

genes or differentially spliced events obtained from the single contrast analyses.
(TIF)

**S2 Fig.** Does response curve of (A,B) Palbociclib treated A375 cell, (C,D) GSK595 treated A375 cell, (E,F) Palbociclib treated CHL1 cell and (G,H) GSK595 treated CHL1 cell.
(TIF)

**S3 Fig.** Upregulated (blue) and downregulated (orange) DEGs between CDK4/6i samples in (A) CHL1 and (B) A375.
(TIF)

**S4 Fig.** Upregulated (blue) and downregulated (orange) DSEs between CDK4/6i samples in (A) CHL1 and (B) A375.
(TIF)

**S5 Fig.** Numbers of up-regulated (blue) and down-regulated (orange) differentially expressed genes at 72hrs of CDK4i and PRMT5 inhibition in (A) CHL1 and (B) A375.
(TIF)

**S6 Fig.** Number of splicing events per gene for (A) 6 days CDK4/6i and (B) 72 hours PRMT5i in CHL1 cells and for (C) 6 days CDK4/6i and (D) 72 hours PRMT5i in A375 cells.
(TIF)

**S7 Fig.** Numbers of up-regulated (blue) and down-regulated (orange) differentially spliced events between 72 hours CDK4/6i, 6 days CDK4/6i and PRMT5 inhibition in (A) CHL1 and (B) A375.
(TIF)

**S8 Fig. PCR products showing the expression patterns of MDM4-FL and MDM4-SL mRNA in CHL1 and A375.**
(TIF)

**S9 Fig. Number of differentially spliced events and differentially spliced genes across treatments with the p-value <0.05 threshold.**
(TIF)

**S10 Fig.** Overlap in differentially spliced genes (DSG) for between72 hours CDK4/6i and PRMT5 inhibition for (A) CHL1 and (B) for A375.
(TIF)

**S11 Fig. Number of selected GO terms related to the cell cycle, translation, DNA repair and splicing in all treatments.**
(TIF)

**S12 Fig. Significance of gene sets detected accommodate all samples in both differentially expressed and spliced genes.**
(TIF)

## Acknowledgments

Computational resources were provided by The University of Melbourne's Research Computing Services, the Petascale Campus Initiative, and the Research Computing Facility at the Peter MacCallum Cancer Centre. LHC and PW received training and organizational support from the Bioinformatics Graduate Program at The University of Melbourne. We thank Anna Small and Jodie Kirkland for assistance with manuscript preparation.

## Author Contributions

**Conceptualization:** Karen E. Sheppard, David L. Goode.

**Data curation:** Shatha Abuhammad, Lydia Rui Jia Lim.

**Formal analysis:** Lok Hang Chan.

**Funding acquisition:** Karen E. Sheppard, David L. Goode.

**Methodology:** Lok Hang Chan, Peihan Wang, Shatha Abuhammad, Joseph Cursons.

**Project administration:** Karen E. Sheppard, David L. Goode.

**Resources:** Shatha Abuhammad, David L. Goode.

**Supervision:** Joseph Cursons, Karen E. Sheppard, David L. Goode.

**Visualization:** Lydia Rui Jia Lim.

**Writing – original draft:** Lok Hang Chan, Karen E. Sheppard, David L. Goode.

**Writing – review & editing:** Lok Hang Chan, Lydia Rui Jia Lim, Karen E. Sheppard, David L. Goode.

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
