## [Decision Letter · Decision Letter 0]

11 May 2023

PONE-D-23-08165PRMT5 and CDK4/6 inhibition result in distinctive patterns of alternative splicing in melanomaPLOS ONE

Dear Dr. Goode,

Thank you for submitting your manuscript to PLOS ONE. After careful consideration, we decided that although your manuscript reports exciting finding,  it requires major revision to fully meet PLOS ONE’s publication criteria as it currently stands. Therefore, we invite you to submit a revised version of the manuscript that addresses the points raised during the review process.

We look forward to receiving your revised manuscript.

Kind regards,

Swati Palit Deb

Academic Editor

PLOS ONE

Journal Requirements:

Reviewers' comments:

Reviewer's Responses to Questions

**Comments to the Author**

1. Is the manuscript technically sound, and do the data support the conclusions?

Reviewer #1: Partly

2. Has the statistical analysis been performed appropriately and rigorously? 

Reviewer #1: N/A

3. Have the authors made all data underlying the findings in their manuscript fully available?

Reviewer #1: No

4. Is the manuscript presented in an intelligible fashion and written in standard English?

Reviewer #1: Yes

5. Review Comments to the Author

Reviewer #1: Comments in the attached file.

6. PLOS authors have the option to publish the peer review history of their article (what does this mean?). If published, this will include your full peer review and any attached files.

Reviewer #1: No

---

## [Author Response · Author response to Decision Letter 0]

20 Aug 2023

We thank our reviewer for the insightful and constructive comments. The manuscript has been revised extensively based on these comments, which includes additional information and reorganization of the results to provide greater clarity. Details of all revisions can be found in ‘ChanEtal_PlosOne_Major_changes_for_resubmission.docx’.

1. What are the biological phenotypes of CHL1 and A375 cells after treatment with CDK4/6i or PRMT5i, or both? Why was a group treated with both inhibitors not included in the study?

The strongest change in biological phenotype observed under CDK4/6 and PRMT5 inhibition in both lines was reduction in proliferation, as described in the first section of the Results (page 16, lines 225-230). We have now provided proliferation curves to support this observation, in the new Supplementary Figure S2.

Examining the effects of the combination of both inhibitors is beyond the scope of this study. Our main intent is to determine the degree of overlap in alternative splicing events induced by CDK4/6 and PRMT5 inhibition to investigate whether CDK4/6 may control splicing independently of PRMT5, which our results indicate it does. Experiments from the Sheppard laboratory indicate the combination further reduces proliferation to a point where it could be difficult to harvest enough cells for RNA-sequencing at a depth that permits robust splicing analysis, creating a practical hurdle as well.

2. Which differentially expressed genes and alternatively spliced genes, as well as their associated Gene Ontology terms, were linked to the biological effects of each treatment or the combined treatment? What experimental evidence supports the expression and splicing of these genes and their association with the biological effects of the treatments?

The Gene Ontology term enrichment results presented in Figure 4 show a clear, consistent and strong enrichment for genes involved in the cell cycle for both differentially expressed genes (DEGs) and differentially spliced genes (DSGs). Throughout the manuscript we highlight individual DEGs and DSGs annotated as having well-established links in the literature to the cell cycle, such as CDK12 and E2F family members, as well as MDM2 and MDM4.

3. Was MDM4 expression/splicing identified in RNAseq analyses, considering the known interaction between PRMT5 and MDM4 in response to CDK4/6i?

Yes, skipping of exon 6 in MDM4 was consistently observed in all samples, albeit only at the more permissive threshold of unadjusted p-value < 0.05 in some samples. These results are now shown in new Supplementary Table S4 and supported by PCR amplification of MDM4 in the region containing exon 6 in CHL1 and A375 after CDK4/6 and PRMT5 inhibition (new Supplementary Figure S8). Even at the more relaxed significance threshold of unadjusted p-value < 0.05, a majority of differential splice events are unique to a single treatment (new Supplementary Figure S9).

4. What was the rationale for analyzing the results from CHL1 and A375 cells separately, and how many differentially expressed genes and alternatively spliced genes, as well as their associated Gene Ontology terms, overlapped or were distinct between the two cell lines? Which of these genes and GO terms are linked to the biological effects of the treatments?

Given the heterogeneity in melanoma cell lines, inclusion of results from both CHL1 and A375 were intended to provide biological validation and support the notion that our findings are more broadly applicable to melanoma in general. By reorganizing the Results section of the manuscript as we have, we aimed to make the similarities between the two lines more apparent and easily recognizable. 

As mentioned above, the primary trend in GO terms enrichment is strong representation of cell cycle and division related terms, consistent with the strong effects of each treatment on cell proliferation.

We hope the revisions outlined here satisfy our reviewer’s concerns and improves overall clarity and comprehension for readers.

---

## [Decision Letter · Decision Letter 1]

18 Sep 2023

PRMT5 and CDK4/6 inhibition result in distinctive patterns of alternative splicing in melanoma

PONE-D-23-08165R1

Dear Dr.Goode,

We’re pleased to inform you that your manuscript has been judged scientifically suitable for publication and will be formally accepted for publication once it addresses the reviewer's concern and all outstanding technical requirements.

Kind regards,

Swati Palit Deb

Academic Editor

PLOS ONE

Additional Editor Comments (optional):

Reviewers' comments:

Reviewer's Responses to Questions

**Comments to the Author**

1. If the authors have adequately addressed your comments raised in a previous round of review and you feel that this manuscript is now acceptable for publication, you may indicate that here to bypass the “Comments to the Author” section, enter your conflict of interest statement in the “Confidential to Editor” section, and submit your "Accept" recommendation.

Reviewer #1: All comments have been addressed

2. Is the manuscript technically sound, and do the data support the conclusions?

Reviewer #1: Partly

3. Has the statistical analysis been performed appropriately and rigorously? 

Reviewer #1: Yes

4. Have the authors made all data underlying the findings in their manuscript fully available?

Reviewer #1: Yes

5. Is the manuscript presented in an intelligible fashion and written in standard English?

Reviewer #1: Yes

6. Review Comments to the Author

Reviewer #1: In the revised version, the authors answered and responded to all questions. However, a statement in the abstract is not supported by data. What is the evidence supporting the implication of current data in developing treatment strategies for preventing the emergence of CDK4/6i resistance? There is no evidence related to CDK4/6i resistance, or a combination of CDK4/6i and PRMT5i that are known to be effective to control CDK4/6i resistance. This part needs to be re-stated.

7. PLOS authors have the option to publish the peer review history of their article (what does this mean?). If published, this will include your full peer review and any attached files.

Reviewer #1: No

---

## [Editor Report · Acceptance letter]

25 Oct 2023

PONE-D-23-08165R1 

PRMT5 and CDK4/6 inhibition result in distinctive patterns of alternative splicing in melanoma 

Dear Dr. Goode:

I'm pleased to inform you that your manuscript has been deemed suitable for publication in PLOS ONE. Congratulations! Your manuscript is now with our production department. 

Kind regards, 

on behalf of

Dr. Swati Palit Deb 

Academic Editor

PLOS ONE